# Preconception risk factors and health care needs of pregnancy-planning women and men with a lifetime history or current mental illness: A nationwide survey

Cindy-Lee Dennis[1,2,3]*, Hilary K. Brown[4,5,6], Sarah Brennenstuhl[1], Simone Vigod[3,6], Ainsley Miller[7], Rita Amiel Castro[8], Flavia Casasanta Marini[2], Catherine Birken[9,10]

1 Lawrence S. Bloomberg Faculty of Nursing, University of Toronto, Toronto, Ontario, Canada, 2 Li Ka Shing Knowledge Institute, St. Michael's Hospital, Toronto, Ontario, Canada, 3 Department of Psychiatry, University of Toronto, Toronto, Ontario, Canada, 4 Interdisciplinary Centre for Health & Society, University of Toronto Scarborough, Toronto, Ontario, Canada, 5 Dalla Lana School of Public Health, University of Toronto, Toronto, Ontario, Canada, 6 Women's College Research Institute, Women's College Hospital, Toronto, Ontario, Canada, 7 School of Nursing, Lakehead University, Thunder Bay, Ontario, Canada, 8 Department of Psychology, University of Zurich, Zurich, Germany, 9 Department of Pediatrics, University of Toronto, Toronto, Ontario, Canada, 10 Hospital for Sick Children, Toronto, Ontario, Canada

* cindylee.dennis@utoronto.ca

## Abstract

### Objectives

While depression and anxiety are common in women and men of reproductive age, preconception interventions to optimize the health of individuals with mental illness before pregnancy is limited and focuses primarily on psychotropic medication management. Comparing individuals with depression, anxiety, and comorbidity to those with neither condition, we identified areas of preconception care optimization related to psychosocial risk factors, general physical health, medication use, and uptake of high-risk health behaviours. We also investigated differences in preconception health care use, attitudes, and knowledge.

### Method

We conducted a nationwide survey of 621 women (n = 529) and men (n = 92) across Canada who were planning a pregnancy within five years, including those with lifetime or current depression (n = 38), anxiety (n = 55), and comorbidity (n = 104) and those without mental illness (n = 413). Individuals with depression, anxiety, and comorbidity were compared to individuals without mental illness using logistic regression, adjusted for age, sex, and education level.

### Results

Individuals with a lifetime or current mental illness were significantly more likely to have several risk factors for suboptimal reproductive and perinatal outcomes, including increased rates of obesity, stress, fatigue, loneliness, number of chronic health conditions, and medication use. Further, they were more likely to have high-risk health behaviours including

**Data Availability Statement:** All relevant data are within the paper and its Supporting Information files.

**Funding:** This work was supported by the Canadian Institutes for Health Research (grant # HLC-154502). The funders had no role in study design, data collection and analysis, decision to publish, or preparation of the manuscript.

**Competing interests:** The authors have no conflicts of interest to declare.

increased substance use, internet addiction, poorer eating habits, and decreased physical activity. By assessing depression, anxiety, or both separately, we also determined there was variation in risk factors by mental illness type.

## Conclusion

Our nationwide study is one of the first and largest to examine the preconception care needs of women and men with a lifetime or current mental illness who are pregnancy-planning. We found this population has many important reproductive and perinatal risk factors that are modifiable via preconception interventions which could have a significant positive impact on their health trajectories and those of their future children.

## Introduction

Preconception health among pregnancy-planning women and men has increasingly been shown to strongly influence reproductive, perinatal, and child outcomes [1]. Yet, many individuals of reproductive age are in poor health at conception, with ever-growing rates of obesity. Since nearly 50% of pregnancies in high-income countries are unplanned [2], risk factors for adverse perinatal outcomes such as poor physical health, medication use, and high-risk health behaviours are often only addressed at the initial prenatal appointment at the end of the critical first trimester [3]. Research into the Developmental Origins of Health and Disease (DOHaD) suggests harmful exposures that occur early in life, while tissues and organs are developing, increase the risk of disease later in life. The DOHaD evidence demonstrates an inextricable link between the in-utero environment and child risk of developing obesity, type 2 diabetes, insulin resistance, asthma, cardiovascular diseases, behavioural disorders, and many other non-communicable diseases [4]. Preconception care is paramount for reducing risk factors and promoting healthy behaviours so that diverse health outcomes can be optimized [5, 6]. While preconception care interventions have been developed for women at high risk for adverse perinatal outcomes, including those with diabetes or HIV [7, 8], minimal attention has focused on the preconception care needs of women and men with common mental illnesses such as depression and anxiety.

Depression and anxiety are major causes of disability worldwide [9]. Lifetime prevalence rates are 20% for depression and 30% for anxiety disorders [9], with higher rates in women than men [10]. Comorbidity is common, as 60% of individuals with anxiety disorders have also experienced a depressive episode [11]. Median age of onset is early in the reproductive years, with a high recurrence rate [9]. Importantly, mental illness in women diagnosed before pregnancy is associated with adverse reproductive and perinatal outcomes [12, 13]. Previous work has also found associations between perinatal mental illness and negative child outcomes, including obesity, asthma, behavioural disorders, mental illness, and many other non-communicable diseases [14–17]. While depression and anxiety have been identified as important conditions in the context of preconception health [18–21], recommendations primarily focus on the safety of pharmacologic treatments in pregnancy—not how to optimize health generally [22]. Preconception care for men with mental health problems is even more limited, primarily highlighting the impact of paternal depression on maternal depression risk and maternal-child interactions [23]. To inform the development of a comprehensive preconception care program that is acceptable, accessible, and scalable for both women and men with common mental health problems, an understanding of their unique preconception needs is required.

## Objectives

The purpose of this nationwide study was to examine the preconception care needs of pregnancy-planning women and men with current or lifetime depression and/or anxiety. Comparing individuals with depression, anxiety, and comorbidity to those with neither condition, our first objective was to identify areas of preconception care optimization in: (1) psychosocial risk factors, (2) general physical health, (3) medication use, and (4) uptake of high-risk health behaviours. Our second objective was to investigate differences in preconception health care use, attitudes, and knowledge.

## Method

### Participants

This study was part of a large cross-sectional survey assessing the preconception care attitudes, beliefs, and intervention preferences of women and men across Canada, conducted in May and June 2019. The study was formative work for Healthy Life Trajectory Initiative (HeLTI) Canada [24]. Participants were recruited via social media, advertisements on provincial public health unit websites, identification of eligible individuals through pre-existing research databases, and referrals from ongoing cohort studies. Women and men were eligible for this study if they were planning to have children within the next five years, could read and understand English, and had access to the Internet. We limited our inclusion criteria to target individuals planning pregnancy in the next 5 years to be consistent with previous studies [25]. Individuals interested in participating received an introductory email after contacting the research team [25]. Written informed consent was obtained from all eligible participants using an online consent form in REDCap [26]; those who provided consent then completed a secure online questionnaire. Research assistants provided reminder follow-up telephone calls to promote completeness of data. Ethics approval was obtained by St. Michael's Hospital (REB# 18–309) and the University of Alberta (REB# 000092107).

### Measures

**Depression and anxiety.** Depression was assessed using two methods. Current depressive symptoms were identified using the Patient Health Questionnaire (PHQ-9) [27], a 9-item scale assessing symptoms in the last 2 weeks. Response options ranged from "not at all" (0) to "nearly every day" (3), with a total score calculated by summing the items. Scores >10 were considered moderate to severe depressive symptoms. This scale is valid and reliable in similar populations [28] and in our sample had a Cronbach's alpha of .83. Lifetime depression was ascertained by, "Have you ever been diagnosed with depression?" Those who replied "yes" and/or scored >10 on the PHQ-9 were coded as positive for ever having depression.

Anxiety was also assessed using two methods. Current anxiety symptoms were identified using the Generalized Anxiety Disorder (GAD-7) [29], a 7-item scale assessing symptoms in the last 2 weeks. Response options ranged from "not at all" (0) to "nearly every day" (3) with a total score calculated by summing the items. A total score of >10 indicated significant anxiety symptoms. This scale is valid and reliable in similar populations [30] and had a Cronbach's alpha of .89 in our sample. Lifetime anxiety was ascertained by, "Have you ever been diagnosed with anxiety?" Those who replied "yes" to this question and/or scored >10 on the GAD-7 were coded as positive for ever having anxiety. Those who were coded positive as ever having depression and anxiety were considered to have comorbid depression and anxiety.

**Psychosocial risk factors.** Three psychosocial risk factors were assessed: perceived stress, fatigue, and loneliness. Stress was measured using the Perceived Stress Scale (PSS) [31]. This

10-item scale asked about frequency of feelings in the past month using response options ranging from 0 to 4. A total score was calculated by summing the items, after reverse-coding four questions. Scores >26 were considered high perceived stress. This scale is valid and reliable in similar populations [32] and in our sample, had a Cronbach's alpha of .87. Fatigue was assessed using, "How fatigued are you during the weekdays?" Response options ranged from "never fatigued" (1) to "almost always fatigued" (5), dichotomized to often or almost always fatigued. Loneliness was assessed using three validated items from the UCLA Loneliness Scale [33] assessing frequency of feeling lack of companionship, left out, and isolated from others. Response options range from "often" (1) to "hardly ever or never" (3). For each item, a binary variable was created that identified those who replied "often".

**General physical health.** Participants were asked to report diagnosed chronic health conditions including: asthma, diabetes, hepatitis B and C, hypertension, inflammatory bowel disease, renal disease, rheumatoid arthritis and other autoimmune diseases, seizure disorders, systemic lupus erythematosus, and thyroid disease. As the prevalence for these conditions was very low (< 3% excluding asthma and thyroid disease), those with 1 or more chronic conditions were grouped. Self-rated health was measured by "How would you rate your overall health?" The response options were: "very healthy", "healthy", "OK", and "unhealthy", dichotomized to identify those who rated themselves as less than healthy (OK/unhealthy) and healthy (healthy/very healthy).

**Medication use.** Medication use was assessed by, "Do you currently use any of the following medications?" Responses were "yes" or "no" to: prescribed medications, over-the-counter medications, herbal or natural medications, weight-loss medications/supplements, and athletic products/supplements. The last two categories were combined due to the low number reporting use of weight-loss medications or supplements.

**High-risk health behaviours.** Eight health behaviours were assessed that may increase the risk of adverse reproductive and perinatal outcomes [34]: (1) frequent alcohol use, (2) any cigarette smoking, (3) cannabis use, (4) low physical activity level, (5) overweight and obesity, (6) poor eating habits, (7) internet addiction and (8) sleeping <6 hours/night. Internet addiction, while not previously identified as a risk factor, was included in an exploratory manner based on emerging evidence suggesting that higher level of device use is associated with maternal depression and poorer relationship satisfaction [35, 36]. Frequent alcohol use was defined as alcohol use more than three times per week, measured by "How often do you drink a beverage containing any alcohol?" taken from the PrimeScreen tool described below. Any cigarette smoking was derived from the following item created by the study team for this survey: "On a typical day, how many cigarettes do you smoke?" Occasional smoking was listed as a response category and included with daily smoking in the analysis. Cannabis use was defined as at least monthly use, derived from an item created by the study team for this survey, "In the past 12 months, have you used cannabis (marijuana) for non-medical/recreational reasons?" For those with a positive response, another question was asked about frequency of use which was used to classified as either used at least once in the past month vs. less. Low physical activity level was defined as obtaining <600 metabolic equivalents per week, based on the Global Physical Activity Questionnaire (GPAQ) [37]. Overweight and obesity were defined as having a BMI ≥25, based on self-reported weight and height. Poor eating habits were defined as having a total score <12 on the PrimeScreen tool, which contains 24 questions about dietary intake [38]. Internet addiction was defined as having a total score >30 on the Internet Addiction Test, which indicates at least a mild/moderate addiction [39]. Finally, sleeping <6 hours/night was based on, "How many hours of sleep do you usually get a night on a weekday/weekend?"

**Preconception health care use, attitudes, and knowledge.** Preconception health care use was measured by "Have you ever received preconception health care from a healthcare

professional?" Response options were "yes" and "no". General attitudes surrounding preconception care were measured using a question designed by the study team, "How important do you consider preconception health for the public, in general?" Responses ranged from "very important" (1) to "not important at all" (5). The variable was dichotomized to identify those with negative attitudes, defined having a score >3. Preconception health knowledge was assessed using the Preconception Health Knowledge Questionnaire (PHKQ), a 25-item multiple choice knowledge test that has been validated in this sample [40]. Responses were coded as being correct or incorrect, and a total score was calculated based on the number of correct responses.

**Demographics.** We also collected data on the demographic characteristics of the sample, including: age (in years), sex (women/men), household income (<$75,000, or ≥$75,000 CAD), education level (high school/college, or university degree), employment status (paid part- or full-time employment, or no paid employment [i.e., on parental leave, unemployed, student, or other]), being born in Canada (yes/no), ethnic origin, and having a child < 2 year of age (yes/no). As pregnancy loss and infertility are related to mental health [41, 42], we reported the proportion of sample that experienced miscarriage and used/were using fertility treatments.

## Data analysis

Sample characteristics were summarized using means and standard deviations for continuous variables and frequencies and percentages for nominal variables and presented by mental illness status. Comparisons across groups were made using an omnibus test from a one-way ANOVA for age (which was approximately normally distributed); Chi square tests were used to compare the distributions of the other sample characteristics. To identify areas of preconception care optimization in individuals with lifetime or current depression, anxiety, and comorbidity, we calculated adjusted odds ratios (aOR) for each group for (1) psychosocial risk factors, (2) general physical health, (3) medication use, and (4) uptake of high-risk health behaviours using logistic regression, with those with neither condition serving as the comparison group. To investigate differences in preconception care use, attitudes, and knowledge, we used logistic regression to calculate aORs for prior use of preconception health care and negative preconception health attitudes, and Ordinary Least Squares regression to calculate the unit-change in preconception health knowledge for each of the comparisons. All models controlled for age, sex, and education level; adjustment variables were selected a priori. All analyses were undertaken using SPSS (v25). Statistical significance was established as $p < .05$. Missing data ranged from model to model, with a high of <10% in models of some health behaviours (e.g., sleep), preconception health knowledge, and attitudes about preconception care.

## Results

### Sample characteristics

Of the 611 pregnancy-planning participants with data on depression and anxiety, 38 (6.4%) had lifetime or current depression, 55 (9.0%) had lifetime or current anxiety, and 104 (17.0%) had lifetime or current depression and anxiety (comorbidity). The mean age of the sample was 32.9 years and ranged from 19 to 60. Women made up 85% of the sample. The majority had a European ethnic origin (72.6%) followed by an Asian ethnic origin (14.2%); other identified origins were reported in very small numbers. About a quarter of the sample had experienced pregnancy loss (26.8%) and just under 10% reported using fertility treatments (7.3%). Table 1 presents the sample demographic characteristics by mental illness status. Significant

**Table 1. Characteristics of individuals with depression, anxiety, and comorbidity compared to those without mental illness.**

| Variable | No depression or anxiety (n = 413) | Depression (n = 39) | Anxiety (n = 55) | Comorbidity (n = 104) | P-value [a] |
|---|---|---|---|---|---|
| Age, mean (SD) | 32.88 (4.16) | 32.64 (4.46) | 33 (5.23) | 32.75 (4.37) | .433 |
| Female, n (%) | 337 (82.2) | 36 (92.3) | 52 (94.5) | 96 (92.3) | .003 |
| Household income < $75K CAD, n (%) | 62 (15.0) | 9 (23.1) | 7 (12.7) | 19 (18.3) | .461 |
| High school/college education, n (%) | 88 (21.3) | 12 (30.8) | 12 (21.8) | 38 (36.5) | .009 |
| No paid employment, n (%) | 122 (29.5) | 19 (48.7) | 22 (40.0) | 41 (39.4) | .021 |
| Born in Canada, n (%) | 321 (77.7) | 31 (79.5) | 49 (89.1) | 98 (94.2) | < .001 |
| European Ethnic Origin, n (%) | 274 (70.1) | 33 (89.2) | 43 (81.1) | 71 (70.3) | .036 |
| Youngest child < 2 years old, n (%) | 135 (32.7) | 15 (38.5) | 21 (38.2) | 47 (45.2) | .113 |

[a] Derived from one-way ANOVA for continuous variables and Chi square tests for categorical variables.

differences in the distribution of sex, education level, employment status, nativity and ethnic origin were found by mental health status (none, depression, anxiety and comorbidity).

## Psychosocial factors

One-third of pregnancy-planning participants had high perceived stress (34.7%) and reported being often or almost always fatigued (34.6%); a significant number of participants also reported often lacking companionship (9.2%), feeling left out (8.2%), or feeling isolated (11.0%). Individuals who had depression, anxiety, or both compared to those with no mental illness were significantly more likely to have high perceived stress, fatigue, and loneliness (Table 2). For high perceived stress, the aORs ranged from 2.67 (aOR = 2.67; 95%CI 1.48–4.83) for anxiety to 8.34 (95%CI 5.06–13.74) for comorbidity. For fatigue, the aORs ranged from 2.08 (95%CI 1.12–3.86) for anxiety to 4.51 (95%CI 2.20–9.24) for depression. For the loneliness, the largest aORs were found for lack of companionship, with an aOR of 6.98 (95% CI 3.35–14.17) for comorbidity. For those with anxiety, the elevated aORs were consistently smaller in magnitude than for those with depression or comorbidity, and the aOR for lack of companionship was non-significant.

## General physical health

Nearly one-quarter of pregnancy-planning participants reported being diagnosed with one or more chronic conditions (23.5%) and rated themselves as being less than healthy (24.3%). Individuals with comorbidity compared to those without mental illness had more than twice the odds of having one or more chronic conditions (aOR 2.54, 95%CI 1.58–4.07) or poor perceived health (aOR 2.44, 95%CI 1.52–3.93) (Table 2). No differences were found for those with depression only or anxiety only compared to those without mental illness.

## Medication use

Prescription medications were used by 33.0% of participants, over-the-counter medications by 20.8%, herbal or natural medications by 9.3%, and weight-loss/athletic products/supplements by 4.7%. Individuals who had depression, anxiety, or both compared to those without mental illness had significantly higher odds of prescription medication use (Table 2), ranging from an aOR of 2.80 (95%CI 1.43–5.9) for depression to 5.37 (95%CI 3.37–8.5) for comorbidity. Those with comorbidity also had significantly higher odds of over-the-counter medication use (aOR 1.70, 95%CI 1.03–2.80). No other findings were statistically significant.

**Table 2.** Adjusted odds of psychosocial risk factors, general physical health indicators, medication use, and high-risk health behaviours among individuals with depression, anxiety, and comorbidity compared to those without mental illness [a].

| Variable | N | Depression | | | | Anxiety | | | | Comorbidity | | | |
|---|---|---|---|---|---|---|---|---|---|---|---|---|---|
| | | Odds Ratio [b] | 95% CI | | P-value | Odds Ratio [b] | 95% CI | | P-value | Odds Ratio [b] | 95% CI | | P-value |
| **Psychosocial Risk Factors** | | | | | | | | | | | | | |
| High perceived stress | 597 | 3.597 | 1.812 | 7.138 | < .001 | 2.674 | 1.480 | 4.830 | 0.001 | 8.336 | 5.057 | 13.743 | < .001 |
| Fatigue | 566 | 4.512 | 2.204 | 9.237 | < .001 | 2.080 | 1.121 | 3.861 | 0.020 | 4.161 | 2.565 | 6.750 | < .001 |
| Often lacking companionship | 601 | 3.770 | 1.544 | 9.206 | 0.004 | 1.231 | 0.407 | 3.725 | 0.712 | 3.227 | 1.665 | 6.255 | 0.001 |
| Often feeling left out | 601 | 4.444 | 1.574 | 12.546 | 0.005 | 3.270 | 1.186 | 9.015 | 0.022 | 6.894 | 3.354 | 14.168 | < .001 |
| Often feeling isolated | 601 | 4.407 | 1.839 | 10.560 | 0.001 | 2.601 | 1.089 | 6.211 | 0.031 | 4.800 | 2.567 | 8.978 | < .001 |
| **General Physical Health** | | | | | | | | | | | | | |
| $\geq$ 1 chronic health conditions | 611 | 1.391 | 0.648 | 2.986 | 0.397 | 1.392 | 0.719 | 2.693 | 0.326 | 2.536 | 1.579 | 4.074 | < .001 |
| Fair or poor self-rated health | 611 | 1.819 | 0.878 | 3.767 | 0.107 | 1.574 | 0.823 | 3.009 | 0.170 | 2.441 | 1.517 | 3.929 | < .001 |
| **Medication Use** | | | | | | | | | | | | | |
| Prescription medications | 611 | 2.799 | 1.427 | 5.491 | 0.003 | 3.184 | 1.781 | 5.691 | < .001 | 5.373 | 3.374 | 8.555 | < .001 |
| Over-the-counter medications | 611 | 0.728 | 0.292 | 1.811 | 0.495 | 1.338 | 0.689 | 2.597 | 0.390 | 1.696 | 1.027 | 2.802 | 0.039 |
| Herbal or natural medications | 611 | 0.491 | 0.112 | 2.142 | 0.344 | 1.611 | 0.692 | 3.750 | 0.853 | 0.847 | 0.392 | 1.859 | 0.689 |
| Weight-loss/athletic products | 611 | 0.635 | 0.082 | 4.93 | 0.664 | 1.41 | 0.395 | 5.033 | 0.597 | 1.513 | 0.572 | 3.999 | 0.3404 |
| **High-Risk Health Behaviours** | | | | | | | | | | | | | |
| Frequent alcohol use | 587 | 1.129 | 0.469 | 2.716 | 0.79 | 1.052 | 0.481 | 2.299 | 0.900 | 1.422 | 0.805 | 2.512 | 0.225 |
| Any cigarette smoking | 582 | -- [c] | -- | -- | -- | 0.969 | 0.271 | 3.459 | 0.961 | 1.161 | 0.491 | 2.745 | 0.733 |
| Monthly cannabis use | 611 | 0.333 | 0.044 | 2.526 | 0.287 | 1.338 | 0.488 | 3.668 | 0.572 | 2.10 | 1.061 | 4.150 | 0.033 |
| Low physical activity level | 577 | 0.836 | 0.420 | 1.666 | 0.611 | 1.134 | 0.627 | 2.052 | 0.677 | 0.930 | 0.589 | 1.469 | 0.755 |
| Overweight or obesity | 594 | 2.728 | 1.274 | 5.541 | 0.010 | 0.896 | 0.497 | 1.615 | 0.715 | 1.654 | 1.037 | 2.638 | 0.035 |
| Poor eating habits | 588 | 1.125 | 0.565 | 2.240 | 0.738 | 1.435 | 0.791 | 2.601 | 0.234 | 1.611 | 1.018 | 2.548 | 0.042 |
| Internet addiction | 568 | 1.173 | 0.530 | 2.598 | 0.694 | 2.086 | 1.091 | 3.989 | 0.026 | 1.731 | 1.044 | 2.869 | 0.033 |
| Sleeping < 6 hours / night | 566 | 1.739 | 0.670 | 4.514 | 0.256 | 0.890 | 0.299 | 2.649 | 0.835 | 1.468 | 0.741 | 2.908 | 0.271 |

[a] Current symptoms and/or lifetime diagnosis.

[b] Adjusted for age, sex, and education level.

[c] Could not be calculated due to the small number of individuals with depression who reported smoking.

## High-risk health behaviours

The prevalence estimates of the high-risk health behaviours were 19.6% for frequent alcohol use, 6.0% for any cigarette smoking, 8.5% for cannabis use, 44.0% for low physical activity level, 54.4% for overweight and obesity, 44.7% for poor eating habits, 24.3% for Internet addiction, and 10.6% for sleeping <6 hours/night. Individuals who had comorbidity, compared to those with no mental illness, were more likely to use cannabis (aOR 2.10, 95%CI 1.06–4.15), be overweight or obese (aOR 1.65, 95%CI 1.04–2.64), have poor eating habits (aOR 1.61, 95%CI 1.02–2.55), and have internet addiction (aOR 1.73, 95%CI 1.04–2.87). Those with depression only were more likely to be overweight or obese (aOR 2.73, 95%CI 1.27–5.84), while those with anxiety only were more likely to have an internet addiction (aOR 2.09, 95%CI 1.09–3.99) (Table 2). No other differences were statistically significant.

## Preconception health care use, attitudes, and knowledge

Two-thirds of participants had not previously received preconception care (66.3%) and one in five considered preconception care to be unimportant (19.3%). The mean number of correct responses on the Preconception Health Knowledge Test was 15.8 (63%) out of 25. Individuals

**Table 3. Preconception health care use, attitudes, and knowledge among individuals with depression, anxiety, and comorbidity compared to those without mental illness [a].**

| Variable | N | Depression | | | Anxiety | | | Comorbidity | | |
|---|---|---|---|---|---|---|---|---|---|---|
| | | Odds Ratio [b] | 95% CI | P-value | Odds Ratio [b] | 95% CI | P-value | Odds Ratio [b] | 95% CI | P-value |
| Ever used preconception health care | 611 | 0.665 | 0.310 1.424 | 0.293 | 0.794 | 0.447 1.475 | 0.465 | 1.608 | 1.018 2.539 | 0.042 |
| Consider preconception care less than important | 561 | 1.058 | 0.437 2.565 | 0.900 | 1.133 | 0.514 2.498 | 0.756 | 0.956 | 0.520 1.755 | 0.883 |
| **Variable** | **N** | **β [b]** | **95% CI** | **P-value** | **β [b]** | **95% CI** | **P-value** | **β [b]** | **95% CI** | **P-value** |
| Preconception health knowledge | 563 | 0.705 | -0.640 2.049 | 0.304 | 1.033 | -0.159 2.225 | 0.089 | 1.607 | 0.703 2.511 | 0.001 |

[a] Current symptoms and/or lifetime diagnosis.

[b] Adjusted for age, sex, and education level.

who had comorbidity were more likely to have received preconception care than those without mental illness (aOR 1.61, 95%CI 1.02–2.54) (Table 3). There were no differences across groups in preconception care attitudes. When looking at preconception health knowledge, those with comorbidity had significantly higher knowledge test scores (on average 1.6 points) compared to those without a mental illness. To explain this finding, we examined each question separately to determine which questions were answered correctly by a greater number of those with comorbidity. A greater proportion of those with comorbidity answered questions correctly on when to see a health care provider after thinking about getting pregnant, intimate partner violence, and what environmental exposures to limit.

## Discussion

Our nationwide study is one of the first and largest to examine the preconception care needs of women and men with a lifetime or current mental illness who are pregnancy-planning. Our results clearly suggest that individuals with a mental illness were significantly more likely to have several important risk factors that lead to suboptimal reproductive and perinatal outcomes [1, 34], including increased stress, fatigue, loneliness, number of chronic health conditions, and medication use. Further, they were more likely to have high-risk health behaviours including poorer eating habits, decreased physical activity, increased substance use, and internet addiction. Importantly, those with a mental illness were significantly more likely to be overweight and obese, a factor related to numerous negative health outcomes to both women and their children [43]. By assessing the preconception health optimization needs of those with depression, anxiety, or both separately, we were also able to determine if there was any variation by mental illness type. We found that, in some cases, needs could be specific to a particular mental illness profile. For example, only comorbidity was associated with certain preconception health risk factors such as poor physical health, over-the-counter medication use, cannabis use, and poor eating habits, whereas high perceived stress, fatigue, and loneliness were found consistently for all mental illness types. Taken together, these findings suggest tailoring interventions by mental illness type might be useful, but that all categories of care optimization should be considered in preconception health care planning for individuals with mental illness.

It is encouraging that attitudes about preconception care did not differ between those with any of the mental health outcomes and those without. This suggests that individuals with a mental health problem may be receptive to engaging with preconception health care services. We found that individuals with comorbidity were more likely to have previously used

preconception care compared to those without mental illness. It is possible that due to the complexity of their mental health problems, individuals with comorbidity have more frequent access to the health care system and consequently greater opportunity to receive preconception health care counselling. Perhaps related to this was the unexpected finding that women and men with comorbidity had greater preconception health knowledge than those without mental illness. However, this was not explained by knowledge about mental illness and pregnancy specifically. The reasons for this finding are unclear, but it is speculated that persons with mental illness may already be targeted in the health care system for preconception health care counseling. However, knowledge does not necessarily translate into behaviour change [44], so interventions to improve preconception health should include more than information. To clinically improve reproductive and perinatal outcomes of pregnancy-planning individuals with mental health problems, work with individuals must be conducted in parallel with population-level efforts, including policies that support mental health.

Our finding that women, those with lower education, and those unemployed were overrepresented among individuals with depression, anxiety, and comorbidity is consistent with research demonstrating social disparities in the prevalence of mental health problems [45, 46]. Our finding of socioeconomic differences between people with and without lifetime or current mental health problems demonstrates that preconception care for individuals with depression and/or anxiety should recognize socioeconomic barriers to preconception care access as well as address these social factors and how they impact reproductive and perinatal health through the tailoring of preconception health care content.

Our findings add to the limited research on preconception health care for individuals with mental health problems. Research to date has focused primarily on preconception care in women, with an emphasis on mental illness screening and psychotropic medication counseling [18–21]. However, research has not addressed the broader preconception health needs of women with pre-existing depression and/or anxiety. Preconception care is not routinely offered to men, and recommendations that exist for men with mental health problems only touch on the impact of paternal depression on maternal depression risk and on the quality of maternal-child interactions [23]. The inclusion of men in our sample is a significant strength, given the growing body of literature demonstrating the contribution of paternal preconception risk factors to reproductive and perinatal outcomes [47]. Further, while the number of men was small compared to women (which did not allow for stratified analysis), we were able to calculate sex-adjusted estimates that can be used to guide future research focused on men specifically. Our results suggest several areas for preconception health optimization in pregnancy-planning women and men with mental health problems, and particularly those with comorbidity, including screening for and addressing risk factors related to psychosocial wellbeing, physical health, medication use, and high-risk health behaviours. These data demonstrate the need for preconception health interventions targeting the broad psychosocial, physical, and behavioural wellbeing of individuals with mental health problems, and not only management of their psychotropic medications.

## Limitations

Limitations include the small proportion of men and those without children. Future research should use more diverse samples, including more men and nulliparous individuals, to replicate these findings. While individuals were recruited, not couples, it is possible that both members of a couple participated in the study. Due to the small proportion of men (and therefore low potential for heterosexual couples) and no one identifying as a same-sex couple, we do not believe this would have affected the model estimation. We did not have information on

response rate, as this was an online survey; we only had the number of completed responses. Another limitation is the lack of objective measures of chronic health conditions, medication use, and health behaviours, with data collected through a cross-sectional, self-reported survey. It is possible that the current mental state of participants affected their subjective responses, resulting in stronger associations than would be found if using more objective measures, such as medical records. Related to this, the measures used to ascertain depression and anxiety can only capture symptoms or subjective recall of a diagnosis and not an actual diagnosis. Thus, a limitation of the work is that diagnostic interviews did not confirm lifetime or current mental health status.

## Conclusion

The results of our nationwide survey of pregnancy-planning women and men in Canada suggest that ever having depression, anxiety, or both is associated with a higher likelihood to have increased perceived stress, fatigue, loneliness, chronic health conditions, prescription medication use, and high-risk health behaviours, all of which are associated with suboptimal reproductive and perinatal outcomes [1, 34]. These data provide evidence of the preconception health management needs of individuals with mental health problems. Given that the risk factors examined herein are modifiable, preconception intervention on these factors could have a significant positive impact on the reproductive and perinatal outcomes of women and men with mental health problems and the wellbeing of their children.

## Supporting information

**S1 Dataset.**
(XLS)

## Author Contributions

**Conceptualization:** Cindy-Lee Dennis, Hilary K. Brown, Sarah Brennenstuhl, Simone Vigod, Flavia Casasanta Marini, Catherine Birken.

**Data curation:** Cindy-Lee Dennis.

**Formal analysis:** Cindy-Lee Dennis, Sarah Brennenstuhl.

**Funding acquisition:** Cindy-Lee Dennis, Catherine Birken.

**Investigation:** Cindy-Lee Dennis.

**Methodology:** Cindy-Lee Dennis, Flavia Casasanta Marini, Catherine Birken.

**Project administration:** Cindy-Lee Dennis, Flavia Casasanta Marini, Catherine Birken.

**Resources:** Cindy-Lee Dennis.

**Supervision:** Cindy-Lee Dennis.

**Validation:** Cindy-Lee Dennis.

**Writing – original draft:** Cindy-Lee Dennis, Hilary K. Brown, Sarah Brennenstuhl, Simone Vigod, Ainsley Miller, Rita Amiel Castro, Flavia Casasanta Marini, Catherine Birken.

**Writing – review & editing:** Cindy-Lee Dennis, Hilary K. Brown, Sarah Brennenstuhl, Simone Vigod, Ainsley Miller, Rita Amiel Castro, Flavia Casasanta Marini, Catherine Birken.

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
