## [Decision Letter · Decision Letter 0]

24 Feb 2022

PONE-D-22-04073Preconception risk factors and health care needs of pregnancy-planning women and men with a lifetime history or current mental illness: A nationwide surveyPLOS ONE

Dear Dr. Dennis,

Thank you for submitting your manuscript to PLOS ONE. After careful consideration, we feel that it has merit but does not fully meet PLOS ONE’s publication criteria as it currently stands. Therefore, we invite you to submit a revised version of the manuscript that addresses the points raised during the review process.

The reviewers concur in regarding this as an interesting and important topic.  Please address the concerns about explaining the analysis details, and respond to the requests for additional analysis to the extent possible.

We look forward to receiving your revised manuscript.

Kind regards,

Emily W. Harville

Academic Editor

PLOS ONE

Journal Requirements:

Reviewers' comments:

Reviewer's Responses to Questions

**Comments to the Author**

1. Is the manuscript technically sound, and do the data support the conclusions?

Reviewer #1: Partly

Reviewer #2: Yes

2. Has the statistical analysis been performed appropriately and rigorously? 

Reviewer #1: Yes

Reviewer #2: I Don't Know

3. Have the authors made all data underlying the findings in their manuscript fully available?

Reviewer #1: Yes

Reviewer #2: Yes

4. Is the manuscript presented in an intelligible fashion and written in standard English?

Reviewer #1: Yes

Reviewer #2: Yes

5. Review Comments to the Author

Reviewer #1: Pg 3, line 33 “preconception care use” is there a word missing here? Not sure what that is.

Pg 5, line 66 there needs a space between “optimized” and in-text citations

Pg 5, I understand the key of this manuscript is to link mental illness and adverse pregnancy outcomes but the first paragraph and the entire introduction focuses on those adverse pregnancy outcomes but leaves a bit to be desired about the focus at hand which is depression/anxiety. I would suggest possibly reframing to make the issue the mental illness and how it relates to other behaviors/medical outcomes

Pg 7, line 101 Could you please explain why the study team chose planning to have children within the next three year as an inclusion criteria. Why three years specifically?

Pg 9, lines 154-159 The questions asked for alcohol, cigarette, and cannabis use, where these taken from other surveys or were these study team created?

Pg 9, line 154 What is the rationale for including internet addiction? I haven’t seen this related to reproductive and perinatal outcomes and am interested in the rational. I reviewed the reference provided for the behaviors and didn’t see internet addiction included in that report so just interested in understanding the link.

Pg 10 lines 169-172 Why only use one question about attitudes? Where did this question come from? Seems that this is pretty important to only rely on one question.

Pg 11 line 204 Could you please provide an actual range distribution for the ages, especially since the lowest mean age is 32.64.

Pg 11 lines 204-206 I think this sentence needs to be worded better. I also think this may be super inflated. The sample sizes are so different so it would make sense.

Table 1 missing the decimal point for “67.9” for Household income for No depression or anxiety

Did you examine any gender differences? I know a major limitation was the lack of having men in your sample…would it make it cleaner to just remove the data from the men?

Also, were these supposed to be couples? Or are the men completely separate from each woman?

Reviewer #2: Authors do not provide enough information to determine if correct statistical analysis was utilized.

Comments for authors

Overall very interesting topic and more data and information is needed on the preconception counseling, support and interventions. Several Comments and suggestions

Line 34 in abstract – I would specifically list the number of women and men

Line 97 – define HeLTI (first use)

Methods

1) Authors do not comment on a survey response rate, this is a key method to a survey study and would recommend including in revisions

2) Authors commend on the Cronbach’s alpha for each survey which is more like a result. I would describe the Cronbach in statistical methods and then place the value in results

3) Line 144 – authors dichotomize to rated themselves as less that healthy – authors should clarify if very healthy, healthy, and ok are considered healthy and unhealthy is considered less than healthy

4) Line 156 – authors define any cigarette smoking by “On a typical day, how many cigarettes do you smoke?”, however a passive smoker that smokes once per week may list 0 – did the survey ask about intermittent smoking and quantify this. Authors should define it as daily smokers, versus intermittent, vs. none

5) Line 157 – again the definition of monthly cannabis seems like assumptions are being made if the questions is in the past 12 months have you used cannabis yes/no someone used it once and answered yes I am not sure I would consider this a monthly user – needs to clarify or define groupings better

6) In data analysis – for ANOVA describe decision making for using one way ANOVA, was normality determined to indicate this is the correct test, in addition what post-hoc analysis test was used to determine significance.

Results

7) Lines 206- 211 include p values for the findings

8) Overall – in all sections authors tend to refer to the table and then just repeat the results in the section. I would suggest so that readers can follow the results better to write is as a summary

Example: Patients with anxiety were more likely to have increased perceived stress (OR, CL); fatigue (OR/CI), feel left out, feel isolated but did not lack companionship. The refer to table for results

9) I know the numbers are small, but I think it would be interesting to stratify by women and men and report the results – many studies don’t include men for preconception evaluation, and I think authors should report what they found on this subset more than what is currently in the paper

General comments

10) Does the survey collect any data on infertility diagnosis or pregnancies failures, were they excluded this is not clear? An infertile couple maybe planning pregnancy in the next 3 years, but mental health may have been affected by unsuccessful attempts at pregnancy. If this information was not collected, I would list as a limitation.

11) Racial breakdown was not included, and I think very important to include as many of the outcomes assessed in this study have racial disparities. If not collected need to list as a limitation.

6. PLOS authors have the option to publish the peer review history of their article (what does this mean?). If published, this will include your full peer review and any attached files.

Reviewer #1: **Yes: **Abigail M. Pauley

Reviewer #2: No

---

## [Author Response · Author response to Decision Letter 0]

1 Apr 2022

Thank you for the opportunity to respond to reviewers’ comments. We have outlined how we have addressed their concerns, which we believe strengthens the manuscript.

Response: We will upload the dataset as requested. 

Reviewer #1:

 Pg 3, line 33 “preconception care use” is there a word missing here? Not sure what that is.

Response: We have added the word health care use to clarify. 

Pg 5, line 66 there needs a space between “optimized” and in-text citations

Response: A space has been added.

Pg 5, I understand the key of this manuscript is to link mental illness and adverse pregnancy outcomes but the first paragraph and the entire introduction focuses on those adverse pregnancy outcomes but leaves a bit to be desired about the focus at hand which is depression/anxiety. I would suggest possibly reframing to make the issue the mental illness and how it relates to other behaviors/medical outcomes

Response: We have further clarified that there is a link between perinatal mental illness various child outcomes: “Previous work has found also associations between perinatal mental health and negative child outcomes, including obesity, asthma, behavioural disorders, mental health illness, and many other non-communicable diseases [14–17].”

Pg 7, line 101 Could you please explain why the study team chose planning to have children within the next three year as an inclusion criteria. Why three years specifically?

Response: We have clarified that “We limited our inclusion criteria to target individuals planning a pregnancy in the next 5 years to be consistent with previous studies [21].” 

Pg 9, lines 154-159 The questions asked for alcohol, cigarette, and cannabis use, where these taken from other surveys or were these study team created?

Response: We have clarified in the measures section that the latter two questions were created by the study team and the former was taken from the PRIMEScreen tool. 

Pg 9, line 154 What is the rationale for including internet addiction? I haven’t seen this related to reproductive and perinatal outcomes and am interested in the rational. I reviewed the reference provided for the behaviors and didn’t see internet addiction included in that report so just interested in understanding the link.

Response: We included internet addiction in an exploratory manner based on emerging evidence that it is associated with maternal depression and poorer relationship satisfaction. We now include this rationale in the measures section. 

Pg 10 lines 169-172 Why only use one question about attitudes? Where did this question come from? Seems that this is pretty important to only rely on one question.

Response: Unfortunately, we only used this general question to investigate attitudes. The question was created by the study team. This is now indicated in the measures section. 

Pg 11 line 204 Could you please provide an actual range distribution for the ages, especially since the lowest mean age is 32.64.

Response: We have provided the mean age for the overall sample and the range, which is large at 19 to 60. Note that the 60-year old is a father. 

Pg 11 lines 204-206 I think this sentence needs to be worded better. I also think this may be super inflated. The sample sizes are so different so it would make sense.

Response: We have reworded the section identified. 

Table 1 missing the decimal point for “67.9” for Household income for No depression or anxiety

Response: We have added the missing decimal point.

Did you examine any gender differences? I know a major limitation was the lack of having men in your sample…would it make it cleaner to just remove the data from the men?

Response: We were unable to examine gender differences due to the small numbers of men as you mentioned. We considered removing the men to provide a cleaner analysis. However, this would further obfuscate of preconception needs of men. By including them in the analysis, not only do we draw attention to the importance of studying men in the context of preconception health, but we can, at very least, provide sex-adjusted estimates that can be used to inform future research focused on men. We have provided more justification for including men despite their small numbers in the discussion. 

Also, were these supposed to be couples? Or are the men completely separate from each woman?

Response: It is possible that some were couples, but this was not intended by the design. We are unable to decern how many were couples, but due to the small proportion of men (and therefore low potential for heterosexual couples) and no one identifying as a same-sex couple, we do not believe this would have impacted the model estimation. That said, we have added this concern to the limitations section.

Reviewer #2:

Overall very interesting topic and more data and information is needed on the preconception counseling, support and interventions. Several Comments and suggestions

Line 34 in abstract – I would specifically list the number of women and men

Response: This has been added as requested. 

Line 97 – define HeLTI (first use)

Response: This has been added.

Methods

1) Authors do not comment on a survey response rate, this is a key method to a survey study and would recommend including in revisions

Response: Unfortunately, we do not have this data as this was an online survey. We only have the number of completed responses.

2) Authors commend on the Cronbach’s alpha for each survey which is more like a result. I would describe the Cronbach in statistical methods and then place the value in results

Response: Thank you for this suggestion. Since it was not an objective of the study to test the internal consistency of the measures, we consider the reported alphas to be characteristics of the measures themselves and not results of the study. 

3) Line 144 – authors dichotomize to rated themselves as less that healthy – authors should clarify if very healthy, healthy, and ok are considered healthy and unhealthy is considered less than healthy

Response: We have clarified that it was spilt into two: healthy (OK/unhealthy) and healthy (healthy/very healthy).

4) Line 156 – authors define any cigarette smoking by “On a typical day, how many cigarettes do you smoke?”, however a passive smoker that smokes once per week may list 0 – did the survey ask about intermittent smoking and quantify this. Authors should define it as daily smokers, versus intermittent, vs. none

Response: We have clarified that the two groups were any cigarette smoking (daily or occasional) vs. none. Intermittent smoking would have been captured in occasional smoking. 

5) Line 157 – again the definition of monthly cannabis seems like assumptions are being made if the questions is in the past 12 months have you used cannabis yes/no someone used it once and answered yes I am not sure I would consider this a monthly user – needs to clarify or define groupings better

Response: We have clarified that “For those with a positive response, another question was asked about frequency of use which was used to classified as either used at least once in the past month vs less.”

6) In data analysis – for ANOVA describe decision making for using one way ANOVA, was normality determined to indicate this is the correct test, in addition what post-hoc analysis test was used to determine significance.

Response: ANOVA was only used for the sample description and only for the variable of age, which was approximately normally distributed. No post hoc tests were undertaken; the p value is from the omnibus test. We have now included these details in the text.

Results

7) Lines 206- 211 include p values for the findings

Response: We have rewritten this section to address another Reviewer’s comment. And, to address your own comment below, we have simplified it greatly and no longer provide specific findings. 

8) Overall – in all sections authors tend to refer to the table and then just repeat the results in the section. I would suggest so that readers can follow the results better to write is as a summary. Example: Patients with anxiety were more likely to have increased perceived stress (OR, CL); fatigue (OR/CI), feel left out, feel isolated but did not lack companionship. The refer to table for results

Response: We have simplified the results when appropriate.

9) I know the numbers are small, but I think it would be interesting to stratify by women and men and report the results – many studies don’t include men for preconception evaluation, and I think authors should report what they found on this subset more than what is currently in the paper

Response: Unfortunately, stratification was not possible due to the small numbers of men with any of the mental health outcomes. 

General comments

10) Does the survey collect any data on infertility diagnosis or pregnancies failures, were they excluded this is not clear? An infertile couple maybe planning pregnancy in the next 3 years, but mental health may have been affected by unsuccessful attempts at pregnancy. If this information was not collected, I would list as a limitation.

Response: Women were included if they were planning for a pregnancy in the next 5 years. Women using fertility treatments and those who had experienced pregnancy loss were not excluded. We now report the proportion of the sample that reported having a miscarriage and using fertility treatments. 

11) Racial breakdown was not included, and I think very important to include as many of the outcomes assessed in this study have racial disparities. If not collected need to list as a limitation.

Response: We did not collect data on race, but now include data on ethnic origin.

---

## [Decision Letter · Decision Letter 1]

9 May 2022

PONE-D-22-04073R1Preconception risk factors and health care needs of pregnancy-planning women and men with a lifetime history or current mental illness: A nationwide surveyPLOS ONE

Dear Dr. Dennis,

Thank you for submitting your manuscript to PLOS ONE. After careful consideration, we feel that it has merit but does not fully meet PLOS ONE’s publication criteria as it currently stands. Therefore, we invite you to submit a revised version of the manuscript that addresses the points raised during the review process.

We look forward to receiving your revised manuscript.

Kind regards,

Emily W. Harville

Academic Editor

PLOS ONE

Journal Requirements:

Additional Editor Comments (if provided):

Please address reviewer 2's minor points.

Reviewers' comments:

Reviewer's Responses to Questions

**Comments to the Author**

1. If the authors have adequately addressed your comments raised in a previous round of review and you feel that this manuscript is now acceptable for publication, you may indicate that here to bypass the “Comments to the Author” section, enter your conflict of interest statement in the “Confidential to Editor” section, and submit your "Accept" recommendation.

Reviewer #1: All comments have been addressed

Reviewer #2: All comments have been addressed

2. Is the manuscript technically sound, and do the data support the conclusions?

Reviewer #1: Yes

Reviewer #2: Yes

3. Has the statistical analysis been performed appropriately and rigorously? 

Reviewer #1: Yes

Reviewer #2: Yes

4. Have the authors made all data underlying the findings in their manuscript fully available?

Reviewer #1: Yes

Reviewer #2: Yes

5. Is the manuscript presented in an intelligible fashion and written in standard English?

Reviewer #1: Yes

Reviewer #2: Yes

6. Review Comments to the Author

Reviewer #1: I greatly appreciate the author's responses and care in addressing them all in the manuscript or within the author replies.

Reviewer #2: Many comments in initial review were addressed

1) Response rate was not addressed because collection methods did not allow for this data collection, this should be added to a limitations of the study

2) Table 2 is very busy with lots of data - I would move the headings to the center and highlight each subsection to make following the data easier for readers

3) Line 376 has inaccurate punctuation

7. PLOS authors have the option to publish the peer review history of their article (what does this mean?). If published, this will include your full peer review and any attached files.

Reviewer #1: No

Reviewer #2: No

---

## [Author Response · Author response to Decision Letter 1]

26 May 2022

Response to the Editor:

1. Please address reviewer 2's minor points.

We have done so, with responses below.

Response to Reviewer 1:

1. I greatly appreciate the author's responses and care in addressing them all in the manuscript or within the author replies.

Thank you for your positive assessment of our revision.

Response to Reviewer 2:

1. Response rate was not addressed because collection methods did not allow for this data collection, this should be added to a limitations of the study

We have added this as a limitation. See Lines 385-386: “We did not have information on response rate, as this was an online survey; we only had the number of completed responses.”

2. Table 2 is very busy with lots of data - I would move the headings to the center and highlight each subsection to make following the data easier for readers

We have made this change.

3. Line 376 has inaccurate punctuation

We have edited the punctuation, as follows: “Further, while the number of men was small compared to women (which did not allow for stratified analysis), we were able to calculate sex-adjusted estimates that can be used to guide future research focused on men specifically.”

---

## [Editor Report · Decision Letter 2]

6 Jun 2022

Preconception risk factors and health care needs of pregnancy-planning women and men with a lifetime history or current mental illness: A nationwide survey

PONE-D-22-04073R2

Dear Dr. Dennis,

We’re pleased to inform you that your manuscript has been judged scientifically suitable for publication and will be formally accepted for publication once it meets all outstanding technical requirements.

Kind regards,

Emily W. Harville

Academic Editor

PLOS ONE
---

## [Editor Report · Acceptance letter]

13 Jun 2022

PONE-D-22-04073R2 

Preconception risk factors and health care needs of pregnancy-planning women and men with a lifetime history or current mental illness: A nationwide survey 

Dear Dr. Dennis:

I'm pleased to inform you that your manuscript has been deemed suitable for publication in PLOS ONE. Congratulations! Your manuscript is now with our production department. 

Kind regards, 

on behalf of

Dr. Emily W. Harville 

Academic Editor

PLOS ONE